

# Extreme events in the Amazon after deforestation

Arim Yoon[1,2], Cathy Hohenegger[1], Jiawei Bao[3], and Lukas Brunner[4]

[1]Max Planck Institute for Meteorology, Bundesstraße 53, 20146 Hamburg
[2]International Max Planck Research School on Earth System Modeling (IMPRS-ESM), Bundesstraße 53, 20146 Hamburg
[3]Institute of Science and Technology Austria, Am Campus 1, 3400 Klosterneuburg, Austria
[4]Research Unit Sustainability and Climate Risk, Center for Earth System Research and Sustainability (CEN), University of Hamburg, Hamburg, Germany

**Correspondence:** Arim Yoon (arim.yoon@mpimet.mpg.de)

**Abstract.** Potential self-perpetuating dieback of the Amazon rain forest has been a topic of concern. The concern is that initial deforestation could critically impair the forest's water recycling capacities, further harming the remaining forest through reduced annual precipitation. Many studies have focused on annual mean precipitation changes, due to its widespread perception as a central control on the Amazon rain forest's stability. However, the impact of deforestation goes beyond changes in the
annual mean precipitation. Yet, global coarse-resolution climate models are not well suited to investigate changes in short-duration and localized events due to their coarse resolution. Here, we circumvent these issues by analyzing a full-deforestation scenario simulated by a global storm-resolving model. We focus on changes in the tail of the hourly distribution of precipitation, temperature, and wind. Hourly precipitation becomes more extreme in the absence of the forest than in an intact forest, with an increased occurrence of both no rain and intense rainfall. These changes are driven by enhanced moisture conver-
gence that strengthens vertical velocity. On average, the near-surface temperature rises significantly by about 3.84 °C, and the daily minimum temperature after deforestation becomes similar to the daily maximum temperature before deforestation. Most human heat stress indicators shift to more severe levels, with implications for health and a significant reduction in work productivity. Finally, the mean 10 m wind speed intensifies by a factor of four, with the 99th percentile wind speed doubling. To summarize, our findings, while based on an idealized case, provide a stark warning of the effects of continuing deforestation
of the Amazon.

## 1  Introduction

The Amazon is home to unparalleled biodiversity and a major carbon sink, making its preservation vital. However, more than 20% of the Amazon forest has already been cleared, and 6% has degraded, with further deforestation expected (RAISG, 2022). As forests shape the local energy balance, the water cycle, and the atmospheric dynamics (Bonan, 2008), their removal will
change environmental conditions, potentially in a way that is unfavorable for forest regrowth. Among environmental conditions, precipitation has been a primary focus. Annual mean precipitation, the most broadly used indicator, is indeed a useful metric for assessing ecosystem structure. For instance, Malhi et al. (2009) used annual precipitation and dry season intensity to classify vegetation types and identify climatic thresholds for vegetation transitions. However, annual mean precipitation can obscure





important details about short-duration precipitation events. In fact, extreme precipitation is often more influential on ecosystem
processes than mean conditions (Heisler-White et al., 2009; Smith, 2011; Thompson et al., 2013). Beyond ecological impacts,
intense precipitation also poses significant challenges to infrastructure and agriculture (Wang et al., 2013; Gao et al., 2018;
Guerreiro et al., 2024; Brown et al., 2020; Fowler et al., 2021). Moreover, forest loss often entails elevated heat stress, and
intense winds can damage the forest and alter forest regrowth (Quine and Gardiner, 2007; Zhan et al., 2017; Kotz et al., 2021).
In our previous study, we investigated the mean annual precipitation response to full Amazon deforestation in a storm-resolving
global climate model (Yoon and Hohenegger, 2025). Unlike previous studies, we found that annual mean precipitation remains
almost unchanged under the deforestation scenario. These findings contradict the classification of the Amazon rainforest as a
climate tipping element. However, the impacts of deforestation go well beyond changes in the annual mean. Thus, the goal of
this study is to investigate changes in extreme precipitation, temperature extremes, and gust winds following complete Amazon
deforestation using the same simulations of Yoon and Hohenegger (2025).

Past studies using coarse-resolution global and regional models, with parameterized convection, have found a reduction in
mean precipitation following deforestation (Nobre et al., 1991; Lejeune et al., 2015; Spracklen and Garcia-Carreras, 2015;
Llopart et al., 2018). To the best of our knowledge, no study has so far investigated the impact of full Amazon deforestation on
precipitation extremes. Extreme precipitation is generated by two essential factors (Johns and Doswell III, 1992; O'Gorman and
Schneider, 2009; Muller et al., 2011; Schumacher and Rasmussen, 2020): the availability of atmospheric moisture (Trenberth
et al., 2003; Lenderink and Attema, 2015), supplied by evapotranspiration and moisture convergence, and the strength of
updrafts (Trenberth et al., 2003; Emori and Brown, 2005; Brown et al., 2020; Loriaux et al., 2017). Regarding the first factor,
there is strong agreement among previous studies that evapotranspiration uniformly decreases after complete deforestation,
leading to a reduction in mean precipitable water (Gedney and Valdes, 2000; Medvigy et al., 2011; Hirota et al., 2011; Pires
and Costa, 2013). However, past studies inconsistently reported both increases and decreases in mean moisture convergence.

Concerning the second factor, strong updrafts, the amount of convective available potential energy (CAPE) is often used
as a proxy for it. Some studies have shown that CAPE decreases after deforestation (Wang et al., 2009; Swann et al., 2015;
Lemes et al., 2023), although none of the studies focused on hourly precipitation. In addition to CAPE, the vertical uplifts over
the Amazon basin are predominantly observed in conjunction with moisture convergence, as indicated by observational data
(Viscardi et al., 2024). Therefore, moisture convergence constitutes a second proxy candidate for the identification of enhanced
updrafts and convection (Crook and Moncrieff, 1988; Tiedtke, 1989; Schaefer and Doswell III, 1980; Davies et al., 2013; King
et al., 2022).

Besides precipitation, forests interact with multiple environmental variables in complex ways. Temperature is one of them,
particularly in tropical forests. Tropical species are adapted to stable climate conditions within a narrow temperature range
(Janzen, 1967; Wright et al., 2009; Perez et al., 2016), making them particularly vulnerable not only to an increase in mean
temperature but also to greater variability. There is a strong consensus that deforestation increases the mean temperature due
to biophysical changes. Reduced evapotranspiration results in more net surface energy being redistributed as sensible heat flux
(Perugini et al., 2017; Duveiller et al., 2018; Butt et al., 2023) and lowered surface roughness length weakens turbulence heat
transport, resulting in near-surface heat accumulation (Baldocchi and Ma, 2013; Winckler et al., 2019). Some studies reported





that deforestation shifts the daily maximum distribution toward higher values and the daily minimum toward lower values,
increasing overall variability (Voldoire and Royer, 2004). To further quantify the effect of temperature changes, especially
on humans, heat stress indicators can be used, derived from meteorological variables (e.g., Morabito et al., 2014; Spangler
et al., 2022; Wang et al., 2009). For instance, Alves de Oliveira et al. (2021) have reported that complete deforestation can
cause the same level of heat stress as several degrees of global warming. Their investigation was based on the wet-bulb global
temperature index, which is used in military training, work safety, and outdoor activities.

The wind is another representative cause of disturbance after deforestation. Frequent damaging winds can prevent a full
regrowth of the forest, as young trees with shallow roots and fragile stems that regrow after deforestation may be more vul-
nerable to stronger winds. Deforestation is expected to increase surface winds by lowering roughness length (Lawrence and
Vandecar, 2015; Sampaio et al., 2007; Spracklen and Garcia-Carreras, 2015; Lejeune et al., 2015) and increasing near-surface
wind due to enhanced land and ocean temperature gradient (Good et al., 2008; Llopart et al., 2018; Mu et al., 2023). Moreover,
potentially stronger downdrafts from convective storms may pose a further threat. However, studies have focused on averaged
features, not on the distribution of changes in hourly wind speed, including changes in downdraft, after deforestation.

Beyond the fact that only a handful of studies have investigated changes in short-duration precipitation, temperature, and
wind events after deforestation over the Amazon, the models used were coarse resolution. Their coarse resolution and their
use of convective parameterizations make them unsuitable to represent fast processes and small scales. Therefore, we use the
same simulations as Yoon and Hohenegger (2025), who used a global storm-resolving model with a 5 km horizontal resolution,
and investigate changes in hourly rainfall, diurnal and seasonal temperature, hourly surface wind speed, and heat stress after
deforestation.

## 2 Methods

We used the ICON-Sapphire simulation presented in Yoon and Hohenegger (2025). The horizontal resolution is 5 km with a
convective parameterization switched off, and a 75 km model top with 90 height levels. The domain is global, and simulations
are run for 3 years. Two simulations are conducted with and without the Amazon forest, prescribing different biophysical pa-
rameters in the land surface model (Table 1). The two simulations are named CTL (without deforestation) and DEF (with com-
plete deforestation). For the analysis presented here, precipitation, temperature, and surface wind are hourly averaged, and we
focus on the Amazon basin (see black contour in Fig. 7a). The moisture convergence is computed from the residuals of the mois-
ture balance equation, including the time tendency of total column water vapor (TCW, Moisture Convergence $= P - E - \frac{d(\text{TCW})}{dt}$
). This is done because the direct calculation of moisture convergence is too inaccurate with the available ourpur frequency. 3D
instantaneous data are saved in 6-hourly intervals.

For the analysis, convective Available Potential Energy (CAPE) and Convective Inhibition (CIN) were computed based on
outputs using the MetPy v1.3.1 Python package May et al. (2022). Parcel ascent was modeled with dry adiabatic lifting to the



**Table 1.** The values for CTL are an average of the grid point values from JSBACH over the Amazon, and the values for DEF are taken from a table in Yoon and Hohenegger (2025), which is based on their values from multiple previous studies.

| Parameters | CTL → DEF |
| --- | --- |
| Albedo | 0.12 → 0.18 |
| Leaf Area Index | 8.40 → 2.70 |
| Vegetation fraction | 0.92 → 0.85 |
| Roughness length (m) | 1.80 → 0.05 |
| Root depth (m) | 1.33 → 0.60 |
| Forest fraction | 0.86 → 0.00 |

level of free convection and pseudo-adiabatic moist ascent thereafter, following the approximations of Bolton (1980). CAPE and CIN are calculated at pressure levels using hourly air temperature and dewpoint temperature starting from the surface.

Daily temperature variability is quantified using day-to-day temperature variation (DTDT) index (Karl et al., 1995), defined as the mean absolute difference in daily mean temperature between successive days ($\delta T = T_{i+1} - T_i$) within a given period (Eq. 1 in Ge et al., 2022).

$$DTDT = \frac{1}{n-1} \sum_{i=1}^{n-1} |T_{i+1} - T_i| \qquad (1)$$

$n$: Total days

To diagnose the impact of deforestation on human discomfort due to changes in temperature and humidity, we use seven heat stress indices as described by Schwingshackl et al. (2021): apparent temperature (AT), NOAA heat index (HI), humidex (Hu), simplified wet-bulb globe temperature ($T_{WBG_s}$), indoor wet-bulb globe temperature ($T_{WBG}$), wet-bulb temperature ($T_{WB}$), universal thermal climate index (UTCI). These indices serve different purposes, leading to a wide range of formulations, with no single index universally regarded as superior (Barnett et al., 2010; Burkart et al., 2011; Schwingshackl et al., 2021). Therefore, to better estimate the impact of deforestation on heat stress, it is required to analyze the overall characteristics of these indices.

HI and Hu are primarily used as heat warning indices. HI, widely applied for assessing heat stress based on temperature and relative humidity, categorizes heat risk into four levels: caution (27 °C, fatigue possible), extreme caution (32 °C, heat stroke, cramps, or exhaustion possible), danger (41 °C, heat stroke, cramps, or exhaustion likely), and extreme danger (54 °C, heat stroke, cramps, or exhaustion highly likely). Hu, developed in Canada, combines temperature and vapor pressure to evaluate thermal discomfort, with threshold indicating some discomfort (30 °C), great discomfort (40 °C), dangerous heat stroke (45 °C), and imminent heat stroke (54 °C). $T_{WB}$ is a physiologically relevant heat stress index that defines the adaptability limits to extreme heat. It represents the lowest temperature an air parcel can reach through evaporative cooling, incorporating temperature, humidity, and pressure. A threshold of 35°C is considered intolerable for humans and likely lethal. $T_{WBG}$ and $T_{WBG_s}$ are widely used for occupational health assessments, as they account for heat stress levels at different work intensities





and rest/work ratios for acclimatized workers. $T_{WBG}$ is a weighted combination of $T_{WB}$ and air temperature, while $T_{WBG_s}$ provides a computationally efficient alternative of $T_{WBG}$ using a linear combination of temperature and vapor pressure. Both indices have the same thresholds, where increasing heat stress requires 25%, 50%, and 75% rest per hour for levels 1 (29 °C), 2 (30.5 °C), and 3 (32 °C), respectively, while level 4 (37°C) indicates conditions where no work is permitted. UTCI and AT are indices designed to assess thermal comfort. UTCI, a model-based index incorporating air temperature, radiant temperature, wind speed, and humidity, is commonly used in studies evaluating heat-related mortality. Here we use the polynomial approximation based on temperature and vapor pressure introduced by Bröde et al. (2012). Its thresholds classify conditions as moderate (26 °C), strong (32 °C), very strong (38 °C), and extreme heat stress (46 °C). AT, derived from temperature and vapor pressure, is commonly used in epidemiological studies to assess heat-related health risks. Its severity levels range from slight discomfort (28 °C) to moderate (32 °C), strong (35 °C), and extreme discomfort (40 °C). A detailed description of the heat stress levels associated with these indices is summarized in Tables S1 and S2 of Schwingshackl et al. (2021).

## 3  Results

### 3.1  Violent Rain

We start by investigating the intensity of hourly precipitation after deforestation. Figure 1a shows the distribution of hourly precipitation in the Amazon basin for both CTL and DEF. Across all three simulation years, the probability of intense hourly precipitation is consistently higher after deforestation, as indicated by the lighter color lines (Fig. 1a). To better visualize changes, the hourly precipitation rates are categorized into five intensity levels based on the WMO classification (2018): "No rain", "light" rain (<0-2.5 $\mathrm{mm\,hr^{-1}}$), "moderate" rain (2.5–10 $\mathrm{mm\,hr^{-1}}$), "heavy" rain (10–50 $\mathrm{mm\,hr^{-1}}$), and "violent" rain (more than 50 $\mathrm{mm\,hr^{-1}}$). Figure 1b illustrates the percentage changes in each category after deforestation. Importantly, the two extreme categories – no rain and violent rain– exhibit a substantial relative increase in frequency after deforestation. No rain almost triples, and violent rain increases by a factor of 1.5. In contrast, light to heavy rainfall remains largely stable, and because light rainfall dominates the frequency of events, the overall mean precipitation remains unchanged as found in Yoon and Hohenegger (2025).



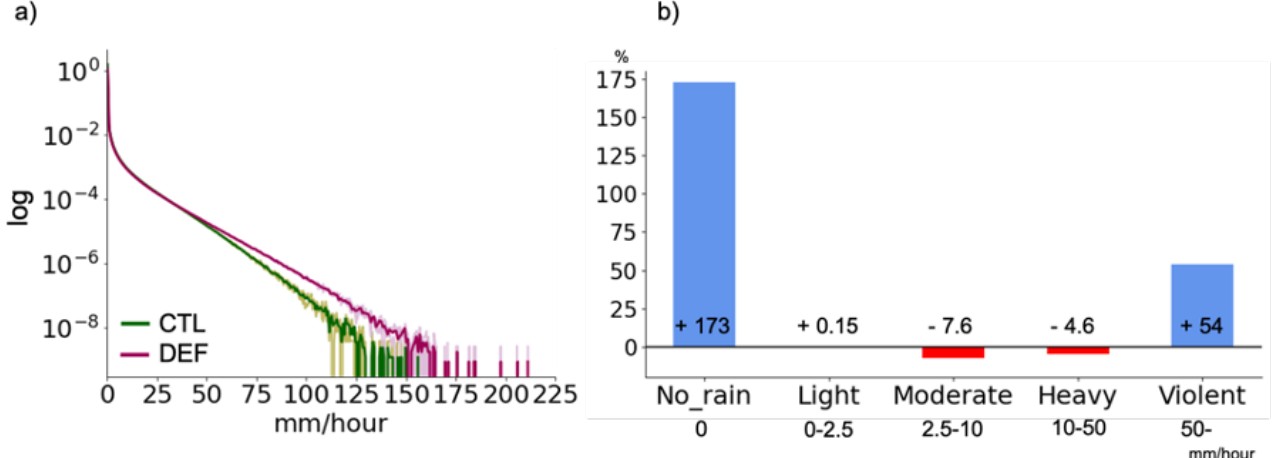

**Figure 1.** Distribution of hourly precipitation $[\mathrm{mm\,hr}^{-1}]$ over the Amazon basin before and after deforestation. (a) The logarithmic probability density function of hourly precipitation rates across all Amazonian grid points. Dark green and dark magenta for CTL and DEF, respectively, using all three years (2020-2022), lighter colors for each year separately. (b) Percentage change (written in numbers, %) in frequency for different intensity categories.

We hypothesize that the changes in the tails of the precipitation distribution can be attributed to the fact that it is more difficult to trigger convection in DEF, leading to more violent outbursts when convection does happen. To confirm this, we
first examine the mechanisms leading to increased violent rain: the availability of atmospheric moisture and strong updrafts (Trenberth, 1999; O'Gorman and Schneider, 2009; Allan and Soden, 2008; Lenderink and Van Meijgaard, 2008; Liu et al., 2009; Muller et al., 2011). Figure 2 shows the intensity of hourly violent precipitation, binned by Total Column Water vapor (TCW) and vertical velocity at 500 hPa ($W_{500}$). Not surprisingly, Figure 2 shows that the intensity of violent rain is stronger with higher TCW and/or stronger $W_{500}$ in both simulations. These links become more pronounced after deforestation, and
more violent rains are simulated in DEF together with higher $W_{500}$ and TCW (Fig. 2b). However, the overall relation between precipitation intensity, TCW and $W_{500}$ remains largely unchanged. More importantly, Figure 2c indicates that the increase in the frequency of violent rain after deforestation comes from a shift towards stronger $W_{500}$, whereas TCW remains in the same range of 50 to 60 mm. Hence, the increase in violent rain is primarily driven by stronger updrafts and not by enhanced TCW (Fig. 2c).




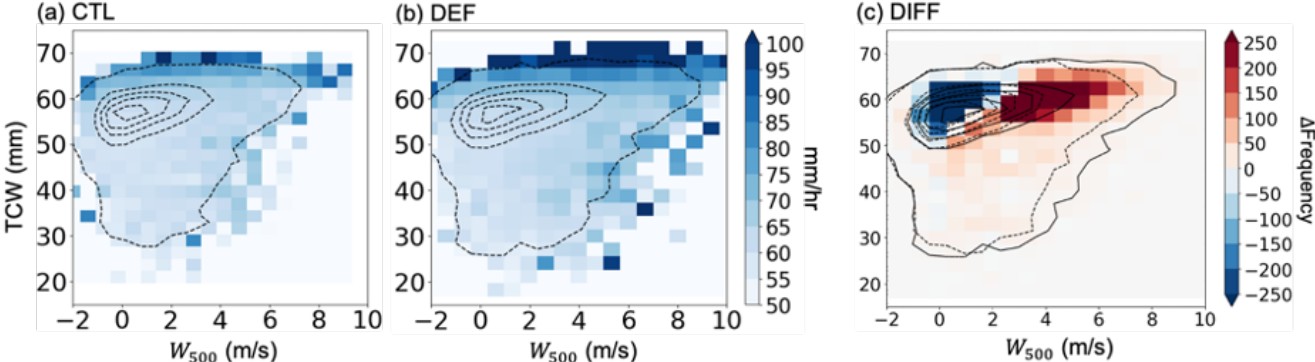

**Figure 2.** The mean intensity of hourly precipitation $[\mathrm{mm\,hr^{-1}}]$ only in violent events, averaged within each bin (shaded) ordered by Total Column Water (TCW) and vertical velocity at 500hPa ($W_{500}$) for (a) CTL and (b) DEF. Dashed contours represent the frequency of TCW and $W_{500}$ occurrences, with contour levels at 100, 500, 1000, 1500, and 2000; the innermost contour corresponds to the highest frequency. (c) Shading shows the difference in frequency (DEF minus CTL) for TCW and $W_{500}$ within each bin. Dashed (CTL) and solid (DEF) lines indicate the frequency distributions already shown in (a) and (b), overlaid for comparison.

Having established that the increase in violent rainfall is mainly due to stronger updrafts, we now investigate the factors responsible for the updraft enhancement. Updraft strength is related to local atmospheric instability and convergence that forces ascent (Davies et al., 2013; Loriaux et al., 2017). Early studies consider moisture convergence as a dynamic variable determined by the circulation (Dai and Trenberth, 2004; Back and Bretherton, 2009). Although moisture convergence mixes the TCW signal, we use it as a proxy for convergence, given that TCW remains unaffected for violent rains after deforestation (Fig. 2c).

First, atmospheric instability is assessed using Convective Available Potential Energy (CAPE). We calculate CAPE one hour prior to the violent rain events in order to relate it to the prerequisites for strong updrafts (Figs. 3). The probability distribution of CAPE values shifts toward lower values after deforestation. The 99th percentile decreases from 3148 $\mathrm{J\,kg^{-1}}$ to 2138 $\mathrm{J\,kg^{-1}}$ (Fig. 3a), and mean CAPE values decrease from 1950 $\mathrm{J\,kg^{-1}}$ to 1058 $\mathrm{J\,kg^{-1}}$. This reduction suggests that the increase in updraft strength is not driven by an increase in local atmospheric instability, as measured by CAPE. To understand why CAPE decreases under deforested conditions, we examine its dependence on near-surface temperature and humidity, as CAPE is directly influenced by both factors (Figs. 3b-c). We take both values from the lowest atmospheric layer to calculate CAPE. While higher temperatures generally increase CAPE, this effect is restricted by the availability of atmospheric moisture. Before deforestation, the highest occurrence of near-surface temperature and humidity is within 24-26 °C and 16-19 $\mathrm{g\,kg^{-1}}$, respectively, where CAPE values range from 1500 to 3000 $\mathrm{J\,kg^{-1}}$. After deforestation, this distribution shifts in the range of 27-30 °C and below 15 $\mathrm{g\,kg^{-1}}$, leading to mean CAPE values around 1000 $\mathrm{J\,kg^{-1}}$. The increase in drier near-surface conditions is a direct consequence of the decrease in evapotranspiration following deforestation, whereas the reduction in CAPE follows from the raised lifting condensation level and level of free convection. In line with our findings, Abramian et al. (2023) also indicated





that CAPE is not a good predictor of the strength of updrafts in their study of squall lines. Hence, increased instability cannot explain the stronger updrafts observed after deforestation.

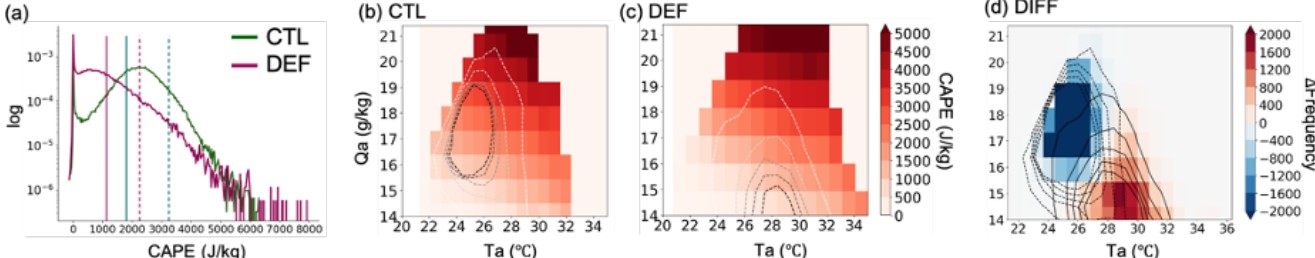

**Figure 3.** CAPE $[\mathrm{J\,kg^{-1}}]$ one hour before violent precipitation over the Amazon. (a) The logarithmic probability density function of CAPE is represented with the mean (solid vertical lines) and the 99th percentile (dashed vertical lines). The PDF is derived from all grid cells and all time steps. CTL is in dark green, and DEF is in dark magenta. CAPE intensity one hour before intense precipitation, binned by near-surface temperature and near-surface specific humidity for (b) CTL and (c) DEF. Colors indicate the average CAPE in each bin. Dashed lines show the frequency of near-surface temperature and specific humidity at levels 100, 500, 1000, 1500, and 2000 times (from white to black). Panel (d) shows the change in the frequency of near-surface temperature and humidity in shading. Dashed (CTL) and solid (DEF) lines indicate the frequency distribution (i.e., the same lines as in Figures b and c).

We now attribute the increase in updraft strength to enhanced moisture convergence. How does moisture convergence change after deforestation, and how does it relate to the occurrence of violent rain events? Figure 4a shows the probability distribution

function of moisture convergence strength one hour before violent precipitation in both CTL and DEF. The tails of the convergence distribution are heavier after deforestation, aligning with the simulated increase in violent precipitation. Now, to answer whether violent rain occurs preferentially in regions with stronger moisture convergence, we examine the spatial patterns of these two variables, separately for dry (Jul-Sep) and wet (Dec-Feb) seasons, given the distinct seasonal circulation patterns in the Amazon region (Marengo, 1992; Leite-Filho et al., 2020; Reboita et al., 2019; Yoon and Hohenegger, 2025). Figures 4b

and c depict regions of stronger moisture convergence in the DEF simulation than in CTL (blue contours in Fig. 4b,c). The frequency change in violent rain is defined as the difference in the number of violent rain occurrences between DEF and CTL for each grid point (shading in Fig. 4b,c). Notably, regions with enhanced convergence largely overlap with those experiencing more violent rains, supporting the hypothesis that enhanced convergence leads to stronger updrafts and, consequently, more violent precipitation. These results are similar to an observation-based study by Davies et al. (2013), which showed a strong

correlation between mechanical updraft and violent precipitation due to moisture convergence rather than CAPE in their study of convective precipitation over the tropical region at Darwin, Australia. Additionally, given that the spatial pattern of stronger moisture convergence aligns with the location of the rainbelt in the corresponding season, violent precipitation appears to be more closely linked to large-scale moisture convergence.



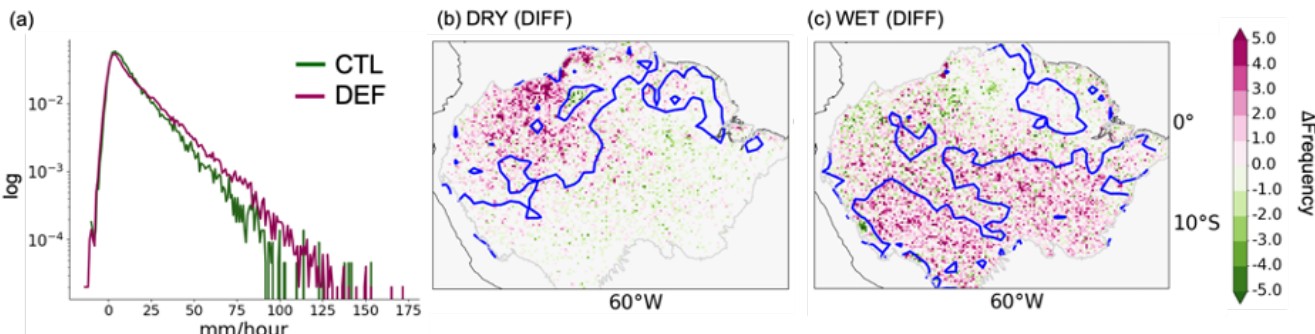

**Figure 4.** (a) Logarithmic probability density function of moisture convergence one hour before violent precipitation events over the Amazon, shown for CTL (dark green) and DEF (dark magenta). (b, c) Differences in the frequency of violent precipitation events between DEF and CTL (DEF minus CTL) at each grid point (shading). The area with a positive anomaly of convergence is defined as the grid points where moisture convergence exceeds the same threshold, 90th percentile of CTL values, and shows a positive anomaly in DEF (Blue contour lines) in (b) the dry season and (c) the wet season.

While our results suggest that increased updrafts are attributed to increased violent rains through convergence, one might wonder why we see more no-rain events. Alongside the decrease in CAPE, Convective Inhibition (CIN) increases, with the mean value rising from $27 \,\mathrm{J\,kg^{-1}}$ to $111 \,\mathrm{J\,kg^{-1}}$ (Fig. 5a). The environment is more inhibited for convection, and this explains why more no-rain events appear. After all, the environment, in general, becomes less favorable to convect thermodynamically, requiring a stronger dynamical driver to precipitate.


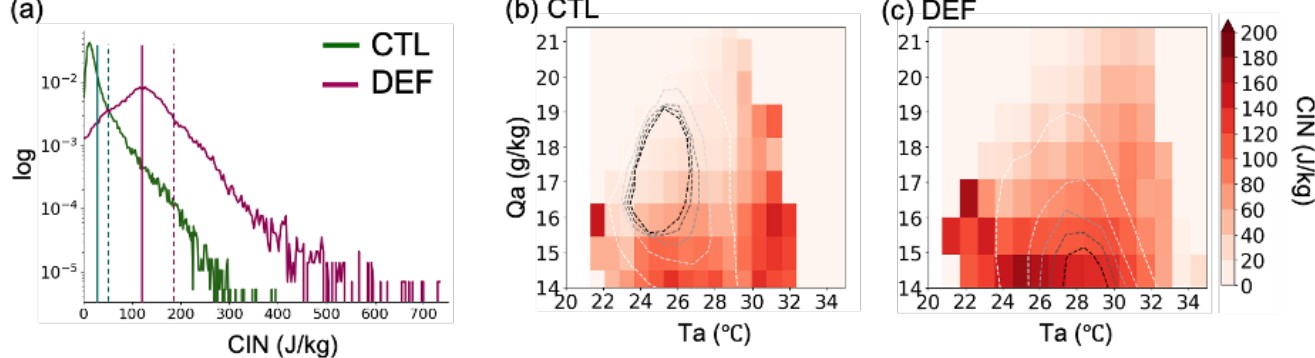

**Figure 5.** Same as Figure 3(a-c) but for CIN [$\mathrm{J\,kg^{-1}}$].





## 3.2 Heat Stress

The mean 2 m temperature increases by 3.84 °C in the annual mean averaged over the Amazon region. Looking at the diurnal
cycle (Fig. 6a), we can see warm temperatures at all times and an increased diurnal temperature range. The post-deforestation
nighttime temperatures become comparable to pre-deforestation daytime values. The temperature changes are significant in
the sense that the difference between DEF and CTL is larger than the interannual variability in temperature in CTL. The tem-
perature distribution of daily mean, daily minimum, and maximum constantly shifts to higher values after deforestation (Fig.
6b)

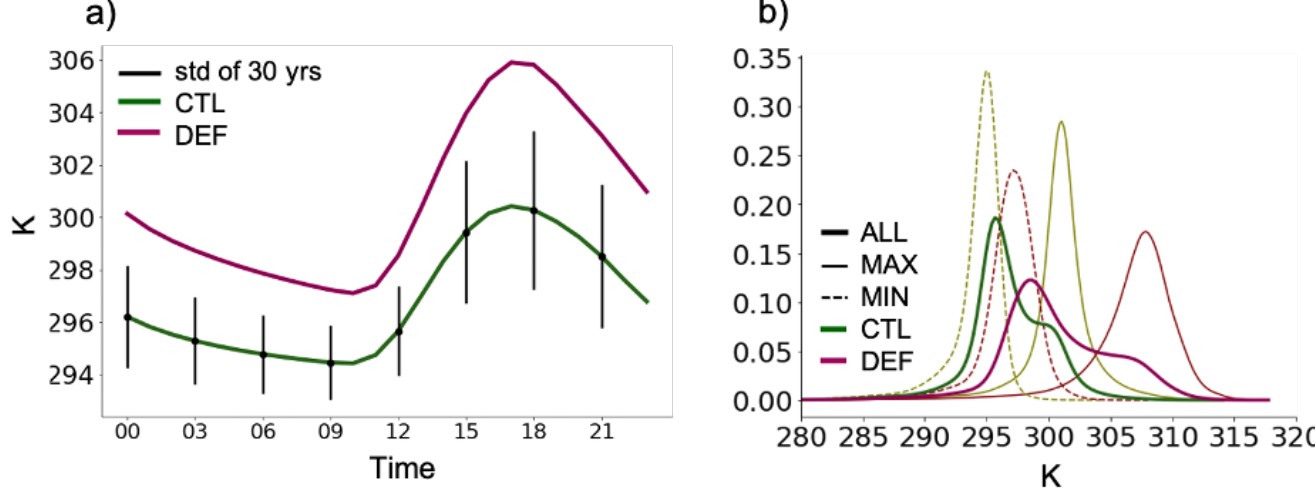

**Figure 6.** Changes in 2 m temperature [K] over the Amazon region after deforestation. (a) The mean diurnal cycle of 2 m temperature in
CTL (dark green) and DEF (dark magenta) with the vertical black lines representing internal variability. The internal variability is computed
from the standard deviations of 2 m temperature from a 30 year CTL simulation conducted with a 10 km grid spacing. (b) Temperature
distributions of the daily mean (thick solid), daily maximum (thin solid), and daily minimum (thin dashed) temperatures for every grid point
in the Amazon.

To understand the temperature changes, we analyze the surface energy budget. The total net surface shortwave radiation
increases due to an increase in downwelling shortwave radiation by 30.93 $\mathrm{W\,m^{-2}}$, which results from reduced overall cloud
cover in DEF. However, the higher albedo partly offsets this increase, reducing it by 16.48 $\mathrm{W\,m^{-2}}$. On the other hand, surface
energy loss by longwave radiation increases from 29.25 $\mathrm{W\,m^{-2}}$ to 61.84 $\mathrm{W\,m^{-2}}$ due to both enhanced upwelling longwave
radiation by warmer surface temperature and reduced downwelling longwave radiation at the surface. Although the combined
effects of shortwave and longwave radiation lead to a net surface radiation decrease of 18.14 $\mathrm{W\,m^{-2}}$, the redistribution of
energy favors sensible heat flux (+38.21 $\mathrm{W\,m^{-2}}$) over latent heat flux (-59.37 $\mathrm{W\,m^{-2}}$), resulting in higher 2 m mean tempera-



ture. This also explains the larger daytime temperature. At night, the temperature in DEF is still higher than the temperature in CTL. This occurs because daytime heating sets a warmer initial condition at the start of nighttime cooling. Although nighttime surface longwave cooling is stronger in DEF, the nighttime period is too short for radiative cooling in DEF to fully offset the temperature difference with CTL. As a result, before DEF can cool to the same extent as CTL, the warming resumes at sunrise,
maintaining a consistently higher 2 m temperature throughout the diurnal cycle. This is consistent with studies that have shown that in the tropical region, open lands tend to be still warmer than forests at night, unlike in boreal regions (Schultz et al., 2017).

We further examine day-to-day variability to understand how temperature fluctuates between days. We use the day-to-day temperature variation (DTDT), which measures the absolute difference in daily mean temperature between consecutive days within a given period (see Methods). Following deforestation, the mean DTDT increases across the Amazon by, on average,
0.4 °C (Fig. 7a), indicating higher day-to-day temperature variability in the deforested case. In Figure 7b, we look at the full distribution of daily $\delta T$ between successive days ($T_{i+1} - T_i$), not just the absolute value between consecutive days (Fig. 7b). The distribution broadens, both on the positive and negative sides. The increase is slightly stronger on the positive side. We find a fourfold increase in the probability of having between -3 and -1 °C changes in temperature between days and an eightfold increase in the probability of having between 1 and 3 °C changes between days. Together with the documented increase in
the diurnal temperature cycle and day-to-day variability, seasonality also increases after deforestation: the range between the yearly maximum and minimum changes from 11.98 to 12.27°C.

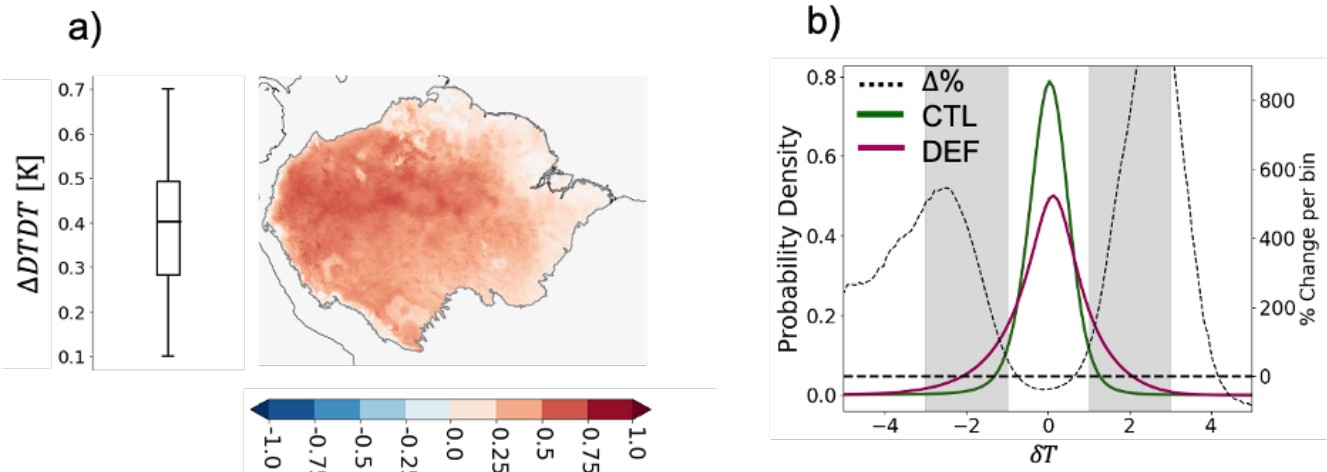

**Figure 7.** (a) Differences in day-to-day temperature variability (DTDT) over the Amazon basin, assessed over the 3-year simulation period. The box-and-whisker plot shows the interquartile range (25th–75th percentiles) as the box, with the horizontal line inside the box indicating the mean. The whiskers extend to the 10th and 90th percentiles. (b) The probability density function of $\delta T$ in CTL (dark green) and DEF (dark magenta) is shown on the left y-axis. Percentage changes between CTL and DEF are represented by the black dashed line on the right y-axis. The grey range is for temperature changes between (-3 K, -1 K) to (1 K, 3 K).



Given these changes, we assess their impact on human thermal stress using seven heat stress indices (see Methods for index explanation). We calculate indices from the full spatio-temporally pooled distribution and show the distribution through the box and whisker plot (Fig. 8). Each index has four heat stress thresholds, represented by color-shading in Figure 8, except $T_{WB}$, which has a single threshold. Although each index categorizes thermal stress levels differently based on its intended application, we standardized the descriptions for AT, HI, Hu, and UTCI as follows: 'level1: slight discomfort', 'level2: moderate discomfort', 'level3: strong discomfort', and 'level4: extreme discomfort'. For $T_{WBG_s}$ and $T_{WBG}$, which are used in occupational health assessments, the levels are defined as 'level1: 25%rest/hour', 'level2: 50%rest/hour', 'level3: 75%rest/hour', and 'level4: 100%rest/hour'. All indices, except $T_{WB}$, consistently indicate a shift toward higher stress levels (Fig. 8). The median AT increases from slight discomfort to strong discomfort. Similarly, the median HI, Hu, and UTCI shift from moderate discomfort to strong discomfort. The $T_{WBG_s}$ suggests that required rest periods increase from 50%rest/hour to 75%rest/hour, indicating a substantial decline in work capacity. Moreover, the number of stressed days increases. In the DEF, 70% of days exceed level 2 across all indices, compared to only 10-30% in CTL. Notably, for level 3, $T_{WBG_s}$ shows a sharp increase, with 63% of days exceeding this threshold in DEF compared to just 13.2% in CTL. In contrast, $T_{WB}$ exhibits a slight reduction after deforestation, as the decrease in humidity offsets the temperature increase. Indoor $T_{WBG}$ increases but does not yet shown any days above level 2 after deforestation, which is not surprising as it is a weighted mean of $T_{WB}$ and near-surface air temperature. Overall, most heat stress indices indicate increased thermal discomfort, higher health risks, and reduced work productivity after deforestation.





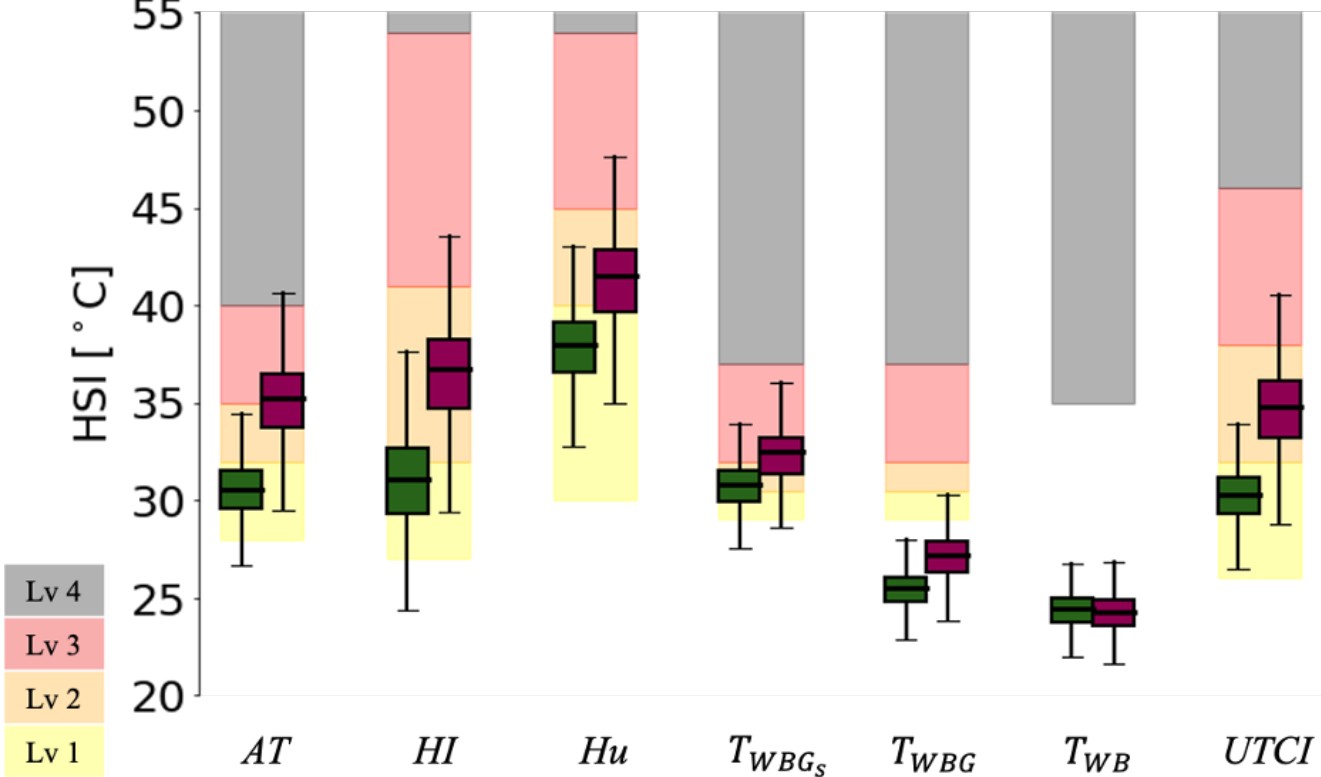

**Figure 8.** Box-and-whisker plots of heat stress indices (HSI), showing the mean, interquartile range (25-75th percentiles), and whiskers (10-90th percentiles): AT, HI, Hu, $T_{WBG_s}$, $T_{WBG}$, $T_{WB}$, UTCI. The background colors show the range of discomfort for each level following Table S1 from Schwingshackl et al. (2021). Yellow shows the range between levels 1 and 2, orange is between 2 and 3, red shows between 3 and 4, and grey is above level 4. Descriptions of indices and levels are given in the Method section.

## 3.3 Damaging Winds

Lastly, we investigate how near-surface wind changes after deforestation. The 10 m wind speed increases from a mean value of $0.93 \, \mathrm{m \, s^{-1}}$ to $3.12 \, \mathrm{m \, s^{-1}}$, and in particular the 99th percentile rises from $3.36 \, \mathrm{m \, s^{-1}}$ to $8.45 \, \mathrm{m \, s^{-1}}$ (shading in Fig. 9). Previous studies have shown that the increase in mean wind speed is a direct result of the decrease in roughness length (Sud et al., 1988) and an intensification of the large-scale circulation after deforestation (Yoon and Hohenegger, 2025). However, beyond that, the increase in hourly precipitation, identified in section 3.1, opens up the possibility of having additional strong winds due to the downdrafts associated with violent rain (Garstang et al., 1998; Windmiller et al., 2023).





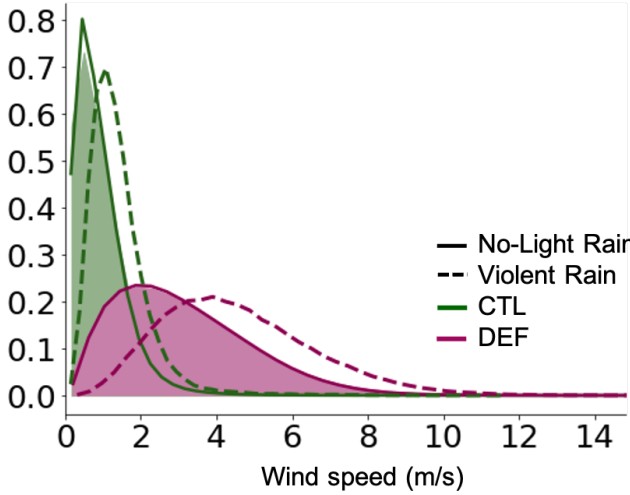

**Figure 9.** Normalized probability density function of 10 m wind speed. Hourly winds from all cases are shaded, with solid lines representing no-light rain cases and dashed lines representing violent rain cases. Wind speeds are collected from all grid cells within the Amazon basin at any time when no-light/ violent rain occurs, and then pooled to construct the probability density function.

We aim to quantify the additional increase in 10 m wind speeds after deforestation that is due to downdrafts associated with
violent rain, separating this effect from changes caused by surface roughness and background circulation. We cannot distinguish between the effect of surface roughness and of background circulation, as we do not have simulations with unchanged roughness at hand. We refer to this factor as R/C and to the downdraft effect as D. To achieve this, we use the Alpert-Stein factor separation method (Stein and Alpert, 1993). We categorize cases into 'no-light rain' (including no rain and light rain) and 'violent rain' in both the CTL and DEF simulations (Table 2). We then assume that wind changes in light rain events primarily
reflect the influence of R/C, whereas changes in violent rain events after deforestation reflect three components: R/C, D, and synergy between R/C and D, see Table 2.

The mean wind speed in CTL during no-light rain is $0.92 \ \mathrm{ms}^{-1}$, whereas it is $3.11 \ \mathrm{ms}^{-1}$ in DEF (solid lines in Fig. 9), indicating a contribution of $2.19 \ \mathrm{ms}^{-1}$ from changes in R/C due to deforestation (f(R/C) in Table 2). In CTL, during violent rain (dashed lines in Fig. 9), the mean wind speed is $1.40 \ \mathrm{ms}^{-1}$. The difference to the no-light rain case is $0.48 \ \mathrm{ms}^{-1}$, and this
represents the contribution from D in an unchanged environment. This comparison indicates that the impact of R/C alone due to deforestation is significantly larger than the impact of downdrafts under control conditions. The wind speed of violent rain is $4.56 \ \mathrm{ms}^{-1}$ in DEF, and it contains both the effect of R/C and of the synergy of R/C and D compared to the violent rain in CTL. This gives a synergy effect of $0.97 \ \mathrm{ms}^{-1}$. In summary, the relative contributions to the total wind speed anomaly are 60% R/C, 13% D, and 27% synergy. This indicates that deforestation amplifies wind speed not only by modifying surface roughness and
circulation but also by strengthening the contribution of downdrafts during violent rain events. Note that the time resolution of outputs, hourly average value, may fail to capture downdrafts, which last for less than 30 minutes (Windmiller et al., 2023).





**Table 2.** The categories of simulations and rain types to disentangle the impact of Roughness length/ background Circulation (R/C), Down-draft (D), and its synergy impact (R/C & D) on windspeed in response to deforestation compared to CTL. The impact of each component is represented with f(0): CTL, f(R/C): roughness length/ background circulation, f(D): downdraft, and f(R/C & D): synergy between R/C and D. Values are in units of $\mathrm{m\,s^{-1}}$. O: included/ X: not included

| Simulation | Rain type | R/C | D | R/C & D | Abbrv. | value |
|---|---|---|---|---|---|---|
| CTL | No-Light rain | X | X | X | f(0) | 0.92 |
| DEF | No-Light rain | O | X | X | f(0)+f(R/C) | 3.11 |
| CTL | Violent rain | X | O | X | f(0)+f(D) | 1.40 |
| DEF | Violent rain | O | O | O | f(0)+f(R/C)+f(D)+f(R/C,D) | 4.56 |

## 4 Conclusions

In this study, we investigated the effect of Amazon deforestation on short-duration events by looking at changes in hourly precipitation, temperature, and winds. We are particularly interested in the changes in the tails of the distributions, given the
threats they pose. To do so, we use global simulations of Amazon deforestation conducted with a grid spacing of 5 km and explicit convection. In contrast to coarse-resolution simulations, such simulations are better suited to investigate changes in extremes.

The main findings are:

– From the five categories of rain events, major changes are only in the tails of the distributions: Violent rain increases by 54% and no-rain by 174%.

– daily minimum and maximum temperatures increase by 2.7 and 5.4°C. Day-to-day temperature becomes more variable, and all heat stress indicators, except for the wet-bulb temperature, point toward higher heat stress.

– the 99th percentile wind values more than doubled.

The increase in violent rains is due to stronger moisture convergence, not stronger CAPE and not stronger TCW, while the increase in no-rain results from increased CIN. The strong warming directly reflects the decrease in evapotranspiration following deforestation, whereas the increase in $T_{WB}$ is mitigated by a strong decrease in near-surface humidity. Finally, we attributed the increase in mean wind speed during violent rain events to changes in roughness length and circulation (60%), downdraft intensification (13%), and synergistic interactions among the two factors (27%). The documented changes
would have impacts on human and forest regrowth. Increased diurnal and seasonal temperature variability will exacerbate the vulnerability of tropical trees, slowing their regeneration. Likewise, the elevated surface wind speeds are expected to create unfavorable conditions for forest regrowth and agriculture. In conclusion, we show that even if annual mean precipitation may remain stable after deforestation, the tail of temperature, precipitation, and wind distribution broadens, making conditions more unfavorable.



*Code availability.* The source code of the ICON model is freely available at http://icon-model.org. The simulations were done with the ICON branch nextgems_cycle2. The IMERG data was downloaded from the Integrated Climate Data Center website (https://www.cen.uni-hamburg.de/en/icdc/data/atmosphere/imerg-precipitation-amount.html). The scripts used to process data and plot the figures in this paper are available in the repository via https://doi.org/10.17617/3.NYLGAN.

*Author contributions.* **Arim Yoon**: conceptualization; data curation; formal analysis; investigation; methodology; software; visualization;
writing – original draft; writing – review and editing. **Cathy Hohenegger**: conceptualization; funding acquisition; investigation; project administration; resources; supervision; review and editing, **Jiawei Bao**: investigation; review and editing, **Lukas Brunner**: visualization; review and editing

*Competing interests.* The authors declare that they have no conflict of interest.

*Acknowledgements.* AY acknowledges funding by the CLICCS centre of excellence subproject A3 funded by DFG. We thank the German
Climate Computing Center DKRZ for providing computing resources and the Integrated Climate Data Center (ICDC), the Center for Earth System Research and Sustainability (CEN), University of Hamburg, for supporting the IMERG data. In addition, we would like to thank Jana Sillmann for suggesting the analysis of heat stress indices and Keno Riechers for providing a thorough internal review of the initial manuscript at the Max Planck Institute for Meteorology. Open Access funding is enabled and organized by Projekt DEAL.



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
