# Peer review of "Extreme events in the Amazon after deforestation"

_EGUsphere, 2025_

## Author Comment (AC1)

**General comments**
**The paper compares the extreme events between two simulations, a control and an Amazon deforestation, using the ICON global model at 5-km resolution. The simulations are based on the two three-year runs, using climatological sea surface temperature. It is an interesting article, well-written and well-organized. However, several conclusions should be drawn more carefully. It is not clear that the CTL run is producing a realistic Amazon climate; validation at the local scale with direct model output variables would give a clearer picture. The unchanged total precipitation with deforestation has implications on the precipitation recycling topic, which requires more attention.**

We thank the reviewer for his/her careful reading of our manuscript and valuable comments. As this work builds on the work by Yoon and Hohenegger (2025), some points mentioned by the reviewer had already been addressed in that study, especially concerning validations, root depth values, and simulation period. But we agree that we should have been more explicit on those points, and we will do so in the revised version. We also expanded the validation of Yoon and Hohenegger (2025) to the specific variables considered in this study, especially extreme precipitation, 2-m temperature, wind, and a simple measure of recycling ratio, using local Fluxnet and global datasets. Concerning the recycling topic, we agree that the unchanged total precipitation has implications for the precipitation recycling topic, as more moisture is supplied from moisture convergence after deforestation. This would mean a reduction in recycling. Unfortunately, we don't have the output that would be required to really trace backwards the origin of the moisture that is falling as precipitation over the Amazon, so we cannot say at the outset how much recycling is changed. Please find below our detailed response, with the reviewer's comments.

**Specific comments**

**Major concerns about the model setup**

1. **The result of no change to the total precipitation after deforestation should be treated with care.**
**As models increase horizontal resolution and switch off the convective parameterization, convective mixing is not treated within its timescale, and stronger updrafts are produced at the grid scale. This lifting of the air by the updraughts leads more easily to air saturation. The cloud microphysics produces a lot of rain due to saturation, but it does not treat column mixing. If some convective mixing is allowed, those extreme precipitation events should probably decrease.**

[Figure]

Figure R1. Distribution of hourly precipitation [mm/hour] over the Amazon basin from IMERG (grey) and ICON-CTL (black) regridded into a resolution of 0.1°.

The reviewer's comment suggests that improper convective mixing in our simulations with explicit convection should lead to too intense precipitation extremes. To check this, we validated ICON's hourly precipitation distribution against observations (see Fig. R1). ICON has a grid spacing of 5 km, whereas IMERG has a resolution of 0.1 degrees. We regridded ICON onto the IMERG grid using an area-weighted interpolation and computed Amazon-wide normalized PDFs. Figure R1 shows that ICON doesn't overshoot the hourly precipitation tail compared to IMERG. In contrast, models with parameterised convection are known to strongly underestimate extreme precipitation, which has been shown repeatedly in past studies (e.g., Kendon et al., 2017; Prein et al., 2015). We note that ICON slightly overestimates the frequency of violent and heavy rain (0.75% against 0.45% in IMERG), which is compensated by a corresponding slight underestimation of the remaining rain categories, but the differences are small, also given the uncertainties associated with the IMERG dataset. We will add this discussion to the manuscript in section 3.1.

**2. Local validation of control run. Switching off convective parameterization completely may need additional verification at the local scale. I recommend including maps of precipitation over the Amazon region to validate precipitation, temperature, and winds at the local scale.**

We agree that the submitted manuscript lacked a concise model-validation summary. Because this study builds upon Yoon & Hohenegger (2025), which was published and where the CTL simulation was validated, we will add a summary of their findings in the revised version. Those can be seen in Yoon and Hohenegger (2025), their Figure S2b and S3. The CTL simulation reproduces the spatial distribution of precipitation and the seasonal migration of the rain belt extremely well (Fig. S3). The amount of precipitation is also well captured within the Amazon, as the vast area has the difference in precipitation smaller than the standard deviation of IMERG (white area from Fig. S2b). The hourly precipitation is also reasonably well reproduced according to Figure R1.

[Figure]

Figure R2. (a) ICON 2m-temperature anomaly (K) to ERA5. (b) Diurnal cycle of 2 m temperature averaged over the Amazon from ERA5 and ICON, and (c) temperature distribution of the daily (thick solid), daily maximum (thin solid), and daily minimum (thin dashed) temperature for those gridpoints from ERA5 (navy) and ICON (grey) datasets. The same distribution from (d, f) averaged from two Fluxnet sites (BR-Sa1: 2.8567°S, 54.9589°W; BR-Sa3: 3.0180°S, 54.9714°W) with a black line and interpolated nearest gridpoint from ICON-CTL with a grey line.

To validate temperature, we use ERA5 2m temperature, and two Fluxnet observation datasets (BR-Sa1: 2.8567°S, 54.9589°W; BR-Sa3: 3.0180°S, 54.9714°W). ICON has a spatially consistent and uniform cold bias over the Amazon (Fig. R2a). It is a well-known feature of the ICON-Sapphire

model (Hohenegger et al. 2023, Segura et al. 2025), which has persisted across model versions. Also, when looking at the diurnal cycle, the 2-m temperature is systematically about 2 K too cold. Finally, Figures R2c and f show the PDF of 2-m temperatures, the daily maximum temperature, and the daily minimum temperatures. Again, we see the systematic too cold temperatures, whereas the bias disappears when looking at the daily range (maximum-minimum temperature) due to compensation of the systematic bias. Hence, as the bias in 2-m temperature is a systematic model bias and as we are interested in the difference between DEF and CTL, we believe that the effect of this bias on the results is minimal.

[Figure]

[Figure]

Figure R3. (a) Normalized probability density function of hourly 10 m wind speed from ERA5 with black lines and from ICON-CTL with grey lines. (b) Wind speed anomaly between ERA5 and ICON.

The 10 m surface wind is validated against ERA5 hourly data, which also has hourly temporal resolution available and values at 10 m (see Figure R3). In Figure R3a, ICON shows a higher probability of both weak and strong near-surface winds compared to ERA5. When averaged over the simulation period, ICON produces, on average, stronger winds over the Amazon, with a mean RMS of 1.68. This is likely related to the model's fine resolution of surface heterogeneity, which enhances wind contrasts, particularly between the ocean and the land.

In summary, CTL realistically captures the spatial–seasonal pattern and hourly statistics of Amazon precipitation when evaluated against IMERG, whereas near-surface temperature shows a systematic cold bias. Wind is slightly stronger compared to ERA5. All in all, we conclude that CTL reproduces reasonably well the climate over the Amazon. These biases are systematic across experiments and thus do not affect our conclusions, which rely on differences between simulations subject to the same issue. We document these biases in the manuscript. We will add this discussion to the updated version of the manuscript, although we will not add the validation Figures, as the paper is about the response to deforestation.

**3. Forest parameters: Rooting depth is too shallow; Amazon forest roots are much deeper and should sustain evapotranspiration during dry periods.**

The used value of the rooting depth for the forest over Amazon (in the CTL simulation) is the default value that JSBACH is using, and this value has been used in other climate simulations. We are aware that the value is too small compared to the value of 4.0 m given in Gandu et al. (2004). This is also why, for the deforested case, we took a value of 0.05 m (from Correia et al. 2008) and not 1 m as given in Gandu et al. (2004), and reported in Yoon and Hohenegger (2025).

**4. Integration length: Three-year length is short for the runs to reach a stable climatic condition. The 15-day spin-up time to reach climatic conditions is also short.**

Yoon and Hohenegger (2025) already discussed the short integration time period and the evolution of soil moisture, see their discussion in the method section and corresponding Fig. S1. We will add a corresponding sentence referring back to the findings of Yoon and Hohenegger (2025) in the method section.

**5. Validations: The work requires validation at the local scale of direct output variables. Differences and statistics are not enough to show the realism of the simulations (precipitation, temperature, evapotranspiration, winds, in different seasons) in the CTL run. There is not enough discussion on precipitation recycling. This is a major topic.**

[Figure]

Figure R4. Monthly ratio between evapotranspiration and precipitation from two sites of FluxNet observation, averaged (black triangle), and the corresponding area from ICON (grey circles).

Please see answer 2 for the validation part.

About precipitation recycling, we did not attempt a quantitative recycling estimate because reliable diagnostics typically require tagged tracers or Lagrangian/Eulerian moisture-tracking (as in the references suggested by the reviewer below), which is beyond the scope of this study (and we do not have the required output). Instead, as a simple proxy, we computed the ratio between evapotranspiration and precipitation (Fig. R4). First of all, we validated the ratio of evapotranspiration and precipitation simulated in ICON against the FluxNet dataset. Except for July, September, and October, the simulated values agree well with the observations, and the seasonal cycle is generally well reproduced. Throughout the year, ICON slightly overestimates both latent heat and precipitation compared to FluxNet. July is the only month when ICON simulates less precipitation than FluxNet, resulting in a higher ratio. In contrast, because ICON shows a faster recovery from the dry season, precipitation is higher in September and October than in FluxNet, leading to a lower ratio. Second, in DEF, as the mean annual precipitation is maintained but evapotranspiration decreases, the recycling ratio decreases. Still, as we cannot properly diagnose recycling ratios and changes in the mean are not the focus of this paper, we decided not to discuss the recycling ratio in the manuscript.

**6. Concerning Citations that deserve to be mentioned:**
 Thanks for the suggestions. We will add Bottino et al. 2024 to the introduction. In contrast, we won't cite Brito et al (2023) and Pilotto et al (2023) as they are not comparable to our simulation settings/ variables that we want to focus on. Brito et al (2023) investigated the impact of combined warming and deforestation effects, whereas we only consider deforestation. Pilotto et al (2023) implemented historical land use change, only focusing on the southwestern Amazon. Moreover, they didn't examine extreme events, but rather monthly precipitation and average temperature.

- **Works on deforestation in the Amazon, that have carried out analysis on the impacts and extremes.**

**Bottino et al. 2024 (https://doi.org/10.1038/s41598-024-55176-5)**
**Brito et al. 2023 ( https://doi.org/ 10.1002/joc.8158),**
**Pilotto et al. 2023 – (https://doi.org/10.1007/s00382-023-06872-x)**
- **Works on precipitation recycling:**

**Rocha et al. 2017 (http://dx.doi.org/10.1590/0102-77863230006)**
**Salati et al 1979: https://doi.org/10.1029/WR015i005p01250 . Classic paper**

**Technical corrections**
1. **Missing the reference page 1: RAISG, 2022**

Thank you for spotting the missing reference. Will be updated.

2. **Line 190: violent rains can be attributed to increased updraughts, not the other way round.**

Thanks for the correction. We will rephrase the sentence to, "While our results suggest that violent rains can be attributed to increased updrafts through convergence, one might wonder why we see more no-rain events."

3. **Line 63: Typo: not global but globe.**

Thanks for the correction.

4. **Line 86: Typo: not ourpur but output**

Thanks for the correction.

**References**

Yoon, A., & Hohenegger, C. (2025). Muted amazon rainfall response to deforestation in a global storm-resolving model. *Geophysical Research Letters*, *52*(4), e2024GL110503.

Kendon, E. J., Ban, N., Roberts, N. M., Fowler, H. J., Roberts, M. J., Chan, S. C., ... & Wilkinson, J. M. (2017). Do convection-permitting regional climate models improve projections of future precipitation change?. *Bulletin of the American Meteorological Society*, *98*(1), 79-93.

Prein, A. F., Langhans, W., Fosser, G., Ferrone, A., Ban, N., Goergen, K., ... & Leung, R. (2015). A review on regional convection-permitting climate modeling: Demonstrations, prospects, and challenges. *Reviews of geophysics*, *53*(2), 323-361.

Segura, H., Hohenegger, C., Wengel, C., & Stevens, B. (2022). Learning by doing: Seasonal and diurnal features of tropical precipitation in a global-coupled storm-resolving model. *Geophysical Research Letters*, *49*(24), e2022GL101796.

Hohenegger, C., Korn, P., Linardakis, L., Redler, R., Schnur, R., Adamidis, P., ... & Stevens, B. (2023). ICON-Sapphire: simulating the components of the Earth system and their interactions at kilometer and subkilometer scales. *Geoscientific Model Development*, *16*(2), 779-811.

Gandu, A. W., Cohen, J. C. P., & De Souza, J. R. S. (2004). Simulation of deforestation in eastern Amazonia using a high-resolution model. *Theoretical and Applied Climatology*, *78*(1), 123-135.

Correia, F. W. S., Alvalá, R. C. D. S., & Manzi, A. O. (2008). Modeling the impacts of land cover change in Amazonia: a regional climate model (RCM) simulation study. *Theoretical and Applied Climatology*, *93*(3), 225-244.

---

## Author Comment (AC2)

**Review of "Extreme events in the Amazon after deforestation"**
**This study employs a global storm-resolving climate model (ICON-Sapphire, 5 km resolution) to simulate the impacts of complete Amazon deforestation on short-duration extreme events. The model captures precipitation, temperature, and wind extremes more realistically than coarse-resolution models. The authors find that while annual mean precipitation remains largely unchanged, the tails of the distributions shift markedly: violent rainfall and no-rain events increase, heat stress intensifies, and extreme winds strengthen. The analysis attributes violent rainfall increases to enhanced moisture convergence, and stronger winds to both reduced surface roughness and storm downdrafts. Overall, the study concludes that deforestation exacerbates climatic extremes. This is an excellent and important paper that provides novel insights into how Amazon deforestation alters extreme events. I have only a few comments, detailed below.**

We thank the reviewer for his/her careful reading of our manuscript and valuable comments. We agree that some sentences were unclear and will rephrase those to clarify, and we will also add information about biophysical parameters, which were lacking in the submitted version. Please find below our detailed response, with the comments of the reviewer for clarity.

**Major Comments**
**1. At the bottom of page 2, the authors mention that several biophysical changes following deforestation but do not explicitly discuss the role of surface albedo. Since Table 1 shows a notable increase in albedo after deforestation, please clarify how this factor interacts with evapotranspiration and surface energy fluxes in your interpretation.**

We will add a few sentences clarifying the impact of albedo on evapotranspiration and surface energy fluxes at the bottom of page 2. We will add that: Although pasture has a higher albedo that reduces net surface radiation, deforestation shifts the energy partition toward sensible heat as it loses substantial evapotranspiration (Perugini et al., 2017; Duveiller et al., 2018; Butt et al., 2023). The shift in Bowen ratio outweighs the reduction from increased albedo, leading to higher sensible heat flux and higher near-surface temperature. Reduced surface roughness length weakens turbulence heat transport, further contributing to near-surface heat accumulation (Baldocchi and Ma, 2013; Winckler et al., 2019).

**2. Page 4: Please clarify what vegetation or land cover is prescribed after deforestation. Relatedly, explain why the leaf area index is still set to 2.7 rather than 0, despite "complete" deforestation.**

The pasture values are taken from Gandu et al. (2004), who compiled them from multiple earlier studies, which based their values on observations. Gandu et al. (2004) also used a value of 2.7 for leaf area index. We will clarify the source by adding the table number from Yoon and Hohenegger (2025) and adding the reference to Gandu et al. (2004).

**3. Page 7: how about CAPE a few hours earlier? Why was 1 hour selected? Justify why CAPE was calculated only one hour before violent rainfall events. Would the results differ if CAPE were considered several hours earlier**

[Figure]

Figure R1. CAPE [J kg$_{-1}$] (a) two hours and (b) three hours before violent precipitation over the Amazon. The logarithmic probability density function of CAPE is represented with the mean (solid vertical lines) and the 90th percentile (dashed vertical lines). CTL is in dark green and DEF is in dark magenta.

We calculated CAPE one hour prior to violent rainfall events to capture the immediate pre-storm environment. In the tropics, CAPE can vary substantially on sub-hourly to hourly timescales and is rapidly depleted once convection begins (e.g., Sherwood, 1999; Zhang, 2002). By focusing on the 1-h lead, we ensure CAPE reflects the state of the atmosphere just before storm initiation, which aligns with our objective of testing whether violent rainfall occurs in association with enhanced instantaneous instability. We additionally examined CAPE 2 h and 3 h prior to the events (2 h lead time: Fig. R1a; 3 h lead time: Fig. R1b). While the signal weakens with increasing lead time, the results consistently show the same tendency as for the 1 h lead. This gives confidence that our conclusion does not depend sensitively on the exact choice of lead time, although a 1 h primer is the most conservative representation of the pre-storm environment. We will mention in the manuscript that the results do not depend upon the chosen lead time and that we chose 1 hour as the most conservative representation of the pre-storm environment.

**4. On page 10, lines 198–200, the authors state: "The post-deforestation nighttime temperatures become comparable to pre-deforestation daytime values." What does this mean in practice? Please clarify the significance. Do you mean that the nighttime minimum after deforestation is as large as daytime maximum before deforestation? A clearer formulation would help readers interpret the magnitude and implications of this result.**

Yes, this is indeed what we described. We will rephrase accordingly.

**5. Wet bulb temperature is a widely used indicator of heat stress. You could use both temperature and humidity changes to represent heat stress. If the sign change of wet bulb temperature differs from other indices, it is possible that the heat stress change is not significant.**

[Figure]

Figure R2. Box-and-whisker plots of heat stress indices (HSI) are the same as Figure 8 for CTL and DEF simulations (green and magenta), additionally with two sensitive simulations: (i) fixing humidity at the CTL values while allowing temperature to change after deforestation (red boxes), and (ii) fixing temperature at the CTL values while allowing humidity to change (blue boxes).

We agree with the reviewer that Twb shows an insignificant impact of deforestation. To further examine the role of temperature and humidity for each heat stress, we conducted an additional sensitivity test (Fig. R2). We calculated each heat stress by (i) fixing humidity at the control (CTL) values while allowing temperature to change due to deforestation (red boxes), and (ii) fixing temperature at the CTL values while allowing humidity to change (blue boxes). The results show that the strongly increased temperature after deforestation leads to high heat stress in all indices, whereas the decreased humidity after deforestation reduces heat stress. Among all indices, Twb is the only one where the humidity reduction (-1.88 °C) manages to compensate for the increase in temperature (+1.64 °C). This follows from the different formulations of the indices and speaks for using more than one index. We will add this discussion and the Figure in the revised version.

**6. Page 14: I do not understand: In summary, the relative contributions to the total wind speed anomaly are 60% R/C, 13% D, and 27%. Please re-explain the factor separation results more clearly.**

We will rephrase the last two paragraphs of section 3.3 as follows:
We aim to quantify the additional increase in 10 m wind speeds after deforestation that is due to downdrafts associated with violent rain, separating this effect from changes caused by surface roughness and background circulation. We cannot distinguish between the effect of surface roughness and of background circulation, as we do not have simulations with unchanged

roughness at hand. We refer to this factor as R/C and to the downdraft effect as D. To achieve this, we use the Alpert-Stein factor separation method (Stein and Alpert, 1993). We categorize cases into 'no-light rain' (including no rain and light rain) and 'violent rain' in both the CTL and DEF simulations (Table 2). The mean wind during no-light rain in CTL is the baseline case. We then assume that wind changes between no-light rain and violent rain in CTL are due to D. Wind changes in the no-light rain events between CTL and DEF primarily reflect the influence of R/C, whereas wind changes in violent rain events in DEF compared to no-light rain in CTL entail the three components: R/C, D, and synergy between R/C and D.

In CTL, the mean wind speed during no-light rain is 0.92 m/s (see value in Tab. 2). For the violent rain, it is 1.40 m/s. This is an increase of 0.48 m/s, which we attribute to the effect of D alone. By contrast, the mean wind speed for no-light rain in DEF is 3.11 m/s, giving an increase of 2.19 m/s. Hence, the effect of R/C is much larger (f(R/C)=2.19 m/s) compared with D (f(D)=0.48 m/s), showing that R/C dominates the response. In DEF, the mean wind speed during violent rain rises to 4.56 m/s. Compared to the no-light rain in CTL, this is an increase of 3.64 m/s (4.56 – 0.92). Given the contributions of 2.19 m/s for R/C and of 0.48 m/s for D, their synergy account for 0.97 m/s. Expressed in percentage, this gives contributions of 60% from R/C, 13% from D, and 27% from their synergy.

**Minor comments**

**On page 3, please add a map clearly showing the spatial extent of the deforested region in the simulations.**
We agree that a reader would like to know the spatial extent of the deforested region. Since the requested area is shown in Fig. 7a and to avoid redundancy while ensuring clarity, we will explicitly refer to Fig. 7a on page 3.

**References**
Here, we don't mention references that are already cited in my paper.

Gandu, A. W., Cohen, J. C. P., & De Souza, J. R. S. (2004). Simulation of deforestation in eastern Amazonia using a high-resolution model. *Theoretical and Applied Climatology*, *78*(1), 123-135.

Sherwood, S. C., & Wahrlich, R. (1999). Observed evolution of tropical deep convective events and their environment. *Monthly Weather Review*, *127*(8), 1777-1795.

Zhang, G. J. (2002). Convective quasi-equilibrium in midlatitude continental environment and its effect on convective parameterization. *Journal of Geophysical Research: Atmospheres*, *107*(D14), ACL-12.

da Silva, L. M., & Santos da Mota, M. A. (2022). Thermodynamic analysis of convective events that occurred in Belém-PA city. *Atmósfera*, *35*(2), 331-355.